# PROCESS-SUPERVISED REINFORCEMENT LEARNING FOR INTERACTIVE MULTIMODAL TOOL-USE AGENTS

## ABSTRACT

Effective interactive tool use requires agents to master Tool Integrated Reasoning: a complex process involving multi-turn planning and long-context dialogue management. To train agents for this dynamic process, particularly in multimodal contexts, we introduce a sandbox environment for reinforcement learning (RL) that supports tool calling and speech-based user simulation. Our core strategy, Turn-level Adjudicated Reinforcement Learning (TARL), addresses the challenge of credit assignment in long-horizon tasks by employing a Large Language Model (LLM) as a judge to provide turn-level evaluation. To enhance exploration, we integrate a mixed-task training curriculum with mathematical reasoning problems. This unified approach boosts the task pass rate on the text-based $\tau$-BENCH by over 6% compared to strong RL baselines. Moreover, we demonstrate our framework's suitability for fine-tuning a multimodal LLM for agentic tasks. By training a base multimodal LLM on interleaved speech-text rollouts, we equip it with tool-use abilities, paving the way for more natural, voice-driven interactive agents.

## 1 INTRODUCTION

Large Language Models (LLMs) (OpenAI, 2024; AI, 2024; Anthropic.; Team, 2025a; Yang et al., 2025) have demonstrated remarkable understanding and reasoning capabilities across diverse domains. As these models advance, enabling them to interact seamlessly with real-world tools and services has emerged as a promising direction. We aim to create agents that can understand and act upon not just text commands, but also spoken language, which requires a new paradigm for agent training. While interactions can span web interfaces, programming systems, and APIs, the fundamental challenge remains: the agent must interpret complex, often multi-turn user requests and execute appropriate actions, whether the input is typed or spoken.

To tackle this challenge, we focus on interactive tool-use agents. We build upon the experimental setup from $\tau$-BENCH, where an agent assists a simulated user with complex tasks by strategically calling tools. This multi-turn conversational format mirrors real-world applications and presents complex reasoning challenges even for state-of-the-art models. Unlike prior approaches (Prabhakar et al., 2025) that rely on static, pre-collected trajectories, we employ Reinforcement Learning (RL) as our primary training methodology. RL allows agents to learn from dynamic model rollouts in an online manner, which is crucial for handling the variability of real-world interactions.

To support this RL-based training paradigm, we have developed a sandbox environment that facilitates agent interactions with users and tools through API calls using the Model Context Protocol (MCP). A core feature of our infrastructure is its support for both text-based and audio-based user simulation. This allows us to train and evaluate both text-only and multimodal agents, providing a direct path toward our primary goal of developing end-to-end voice agents that can act on spoken commands in realistic scenarios.

However, standard RL algorithms falter in this complex setting. We observed that as training progresses, models often become overconfident, reducing their capacity for exploration. To counteract this, we introduce a two-pronged strategy. First, we employ mixed-task training—incorporating medium-difficulty math problems—to encourage persistent exploration and regularize the learning process. Second, to solve the critical credit assignment challenge in our long multi-turn trajectories, we propose Turn-level Adjudicated Reinforcement Learning (TARL), visualized in Fig. 1.

Figure 1: This illustration outlines our training pipeline for an iterative tool-use agent. The agent operates within a sandbox environment, receiving results from tool executions and feedback from users. We then evaluate and score both individual turns and the complete trajectory, which generates the reward signal used to update the agent.

This method uses an LLM-based judge to provide fine-grained, turn-level rewards that guide policy updates. On text-based tasks, the combination of these techniques boosted the pass rate by an additional 6% over our already strong RL baselines.

Having established our framework's effectiveness in the text domain, we applied it to our main objective: training a multimodal agent with real-world utility. Leveraging our sandbox environment, we trained a base multimodal LLM on $\tau$-BENCH tasks with speech-based user simulation. Guided by our proposed mixed-task training and TARL strategies, our approach successfully equipped the model with robust interactive tool-use abilities, improving the pass rate by over 20% compared to the base model. This demonstrates a viable path for fine-tuning multimodal foundation models for complex agentic tasks using process-supervised RL. In summary, our contributions are threefold:

- A generalizable, open-source sandbox designed for training interactive tool-use agents across both text and speech modalities.
- An enhanced RL training strategy (TARL) that improves performance by encouraging exploration and enabling fine-grained, turn-level credit assignment.
- The first demonstration of this framework to successfully train a multimodal voice agent through RL on interleaved speech-text interactions, showing great performance gains.

## 2 PRELIMINARY

### 2.1 SANDBOX ENVIRONMENT FOR TOOL-USE AGENTS

Our sandbox environment is composed of three integrated components designed for training interactive agents. (1) The backend application uses a relational SQLite database, adapted from the $\tau$-BENCH dataset, and exposes tools to the agent through RESTful APIs. (2) Our user simulator, powered by GPT-4 and SeedTTS, generates text and speech-based user responses. (3) Finally, a rule-based verifier evaluates the agent's actions by comparing its database-altering tool calls against ground-truth data, providing a binary reward to guide reinforcement learning. For more details, please refer to our detailed description of each component in Appendix §A.

### 2.2 RL PRELIMINARIES

We formulate the agent training as a Markov Decision Process (MDP). The policy is an autoregressive language model, $p_\theta$, which generates a sequence of tokens (actions) based on the preceding conversation history (state). An interaction trajectory, $\boldsymbol{\tau}$, is an alternating sequence of agent-generated text, $\boldsymbol{x}^i$, and environment responses, $\boldsymbol{e}^i$. The objective is to learn the policy parameters $\theta$ that maximize the expected trajectory-level reward:

$$J(\theta) = \mathbb{E}_{\boldsymbol{\tau} \sim p_\theta}[R(\boldsymbol{\tau})]$$

where $R(\boldsymbol{\tau})$ is a scalar reward assigned to the entire trajectory. To optimize this objective, we explore a few on-policy RL algorithms.

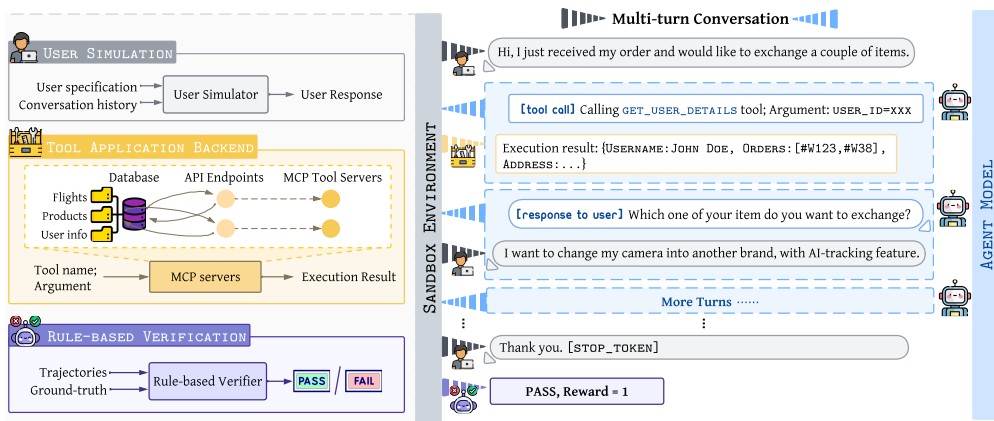

Figure 2: Our environment setup for interactive tool-use agents.

**Proximal Policy Optimization** (Schulman et al., 2017, PPO) is a policy gradient algorithm that stabilizes training by constraining policy updates. It uses a clipped surrogate objective that limits how much the policy can change from one iteration to the next. The advantage function, which measures the relative value of an action, is calculated using Generalized Advantage Estimation (GAE), where all positions share the same reward $R(\boldsymbol{\tau})$, obtained through our rule-based verifier.

**Group Relative Policy Optimization** (Shao et al., 2024, GRPO) enhances PPO by introducing a reward normalization scheme to improve training stability. For a given batch of $G$ trajectories, it calculates the mean ($\mu_R$) and standard deviation ($\sigma_R$) of the rewards. The advantage for a trajectory is then its z-score: $A = (R(\boldsymbol{\tau}) - \mu_R)/\sigma_R$. This normalization makes the training process less sensitive to the scale of rewards.

**REINFORCE Leave-One-Out** (Ahmadian et al., 2024, RLOO) is a variance reduction technique that computes a unique baseline for each trajectory in a batch. The advantage for a specific trajectory $\boldsymbol{\tau}_n$ is its reward minus the average reward of all *other* trajectories in the batch: $A_n = R(\boldsymbol{\tau}_n) - \frac{1}{G-1}\sum_{j \neq n} R(\boldsymbol{\tau}_j)$. This "leave-one-out" baseline is unbiased and effectively reduces the variance of the policy gradient estimates.

For the detailed RL formalization of our multi-turn tool-use setting, please refer to Appendix §B.

## 2.3 BENCHMARK RL ALGORITHMS

After constructing our sandbox environment, we first benchmark RL algorithms on $\tau$-BENCH to understand the capabilities of vanilla RL algorithms on tool-use tasks. We utilize text-based user simulation with Qwen3-8B (Yang et al., 2025) as our base model with training configurations in Appendix §D. For our training data, we use GPT-4.1 to synthesize user instruction prompts and ground-truth tool-call annotations through publicly released trajectory data from APIGEN-MT (Prabhakar et al., 2025)[1]. More details of our data preparation can be found in Appendix §C.

Since there are a very limited number of trajectories and test cases for AIRLINE, we only synthesize RETAIL domain's training data. For evaluation, we assess our models on both RETAIL and AIRLINE domains. Across all our experiments, we use the passˆk metric (Yao et al., 2024) in conjunction with our rule-based verifier. For a given task, passˆk equals 1 only when all k sampled conversation trajectories are verified as correct by the environment.

**In-Domain RL Training Shows Promise but Faces Limitations** Our benchmark results in Table 1 demonstrate that all RL algorithms successfully improve Qwen3-8B's performance on the RETAIL domain. GRPO achieves the largest improvement, closely followed by PPO (both using $n = 4$ rollouts), indicating that RL training effectively enhances the model's tool-using capabilities. The improvement is most pronounced in single-sample scenarios (passˆ1), where GRPO delivers approximately 9% improvement over the baseline.

---

[1] Publicly available at `https://huggingface.co/datasets/Salesforce/APIGen-MT-5k`

| Agent Model | Retail | | | | | | Airline | | | |
|---|---|---|---|---|---|---|---|---|---|---|
| | pass^1 | pass^2 | pass^3 | pass^4 | #Wait | Len | pass^1 | pass^2 | pass^3 | pass^4 |
| *Baseline Models* | | | | | | | | | | |
| GPT-4.1 | 60.9 | 55.7 | 51.3 | 47.8 | 0.3 | 54 | 48 | 34 | 26 | 24 |
| Llama-xLAM-2-8B | 42.6 | 34.8 | 28.7 | 26.1 | 0.1 | 19 | 36 | 26 | 20 | 18 |
| Qwen3-8B | 42.6 | 30.4 | 25.2 | 21.7 | 14.6 | 228 | 32 | 24 | 20 | 20 |
| *Qwen3-8B + RL* | | | | | | | | | | |
| GRPO (n=4) | **51.3** | **37.4** | 30.4 | **27.0** | 11.7 | 204 | 28 | 14 | 8 | 4 |
| RLOO (n=4) | 47.0 | 31.3 | 28.7 | 24.3 | 11.1 | 180 | **36** | 20 | **16** | **12** |
| PPO (n=4) | 48.7 | 36.5 | **31.3** | 26.1 | 8.4 | 162 | 32 | **22** | 14 | **12** |

Table 1: pass^k results of tool-use agents trained with different RL algorithms on $\tau$-BENCH (baseline models–GPT4.1 (OpenAI, 2024), xLAM-2-8B (Prabhakar et al., 2025), and Qwen3-8B (Yang et al., 2025)–are replicated with our environment setup). $n$ denotes the number of rollouts during training. The best RL-trained results are **bolded**. For RETAIL, we also report **#wait** (the average number of "wait" tokens as an indicator of self-reflection) and **Len** (response length per turn).

However, the learned skills do not generalize to out-of-domain AIRLINE tasks, a limitation we attribute to our small, domain-specific training dataset and the fact that AIRLINE tasks are generally harder than the RETAIL domain. Achieving better generalization would require crafting large and diverse environments, as demonstrated by recent work like Kimi-K2 (Team, 2025b). On the other hand, our focus is on optimizing RL strategies for in-domain performance.

**The Confidence Paradox: When More Confidence Isn't Better** While RL training is known to enhance model confidence and sampling efficiency (Shao et al., 2024; Damani et al., 2025; Yue et al., 2025)—indeed reflected in our improved pass^1 results—this increased confidence comes with gradually reduced explorations. Analysis of our sampled trajectories reveals that post-training models exhibit reduced self-reflection and self-correction behaviors, as evidenced by the substantial decrease in "wait" tokens (Qwen3 tends to use phrases like 'wait, ...' to interrupt its thinking process and reflect on its actions) and shorter average response lengths. For example, we observe that the model over-confidently cancels orders without confirming with users, leading to avoidable errors.

Although these behavioral changes do not necessarily translate to lower overall performance, *they significantly impact the exploration benefits of RL training once the model is confidently exploring sub-optimal strategies*. Furthermore, the vanilla use of trajectory-level rewards could be problematic for multi-turn conversations—in our case, with contexts up to 32,768 tokens—as it creates sparse reward signals that lead to suboptimal credit assignment when the model performs multiple actions per trajectory. These challenges inspire us to design training strategies that encourage agent exploration with fine-grained turn-level feedback in the next section.

## 3 METHOD

### 3.1 MIXED-TASK TRAINING

To encourage exploration during training, we propose incorporating medium-difficulty math problems into the training process. This strategy leverages the fact that base models like Qwen3-8B (Yang et al., 2025) have been pre-trained on mathematical and coding problems, giving them strong reasoning capabilities. When solving math problems, language models naturally engage in self-reflection and make multiple self-corrections, which elongates their chain-of-thought (Wei et al., 2023, COT) reasoning trajectories and promotes exploratory behavior. By mixing math problems with RETAIL domain tasks, we regularize the training process to prevent the model from overfitting to the retail domain while preserving its exploration abilities through self-reflection.

In practice, we evaluated several math datasets including GSM8K (Cobbe et al., 2021), DeepScaleR (Luo et al., 2025), and DAPO-MATH-17K (Yu et al., 2025), ultimately selecting medium-difficulty problems from DeepScaleR. We chose this dataset because medium-level problems provide sufficient challenge to force the model to reflect on its reasoning process and generate longer CoT trajectories, while remaining manageable difficulty for an 8B parameter model.

Figure 3: The judge assesses each turn based on the full conversation history with ground-truth annotations. Here, the agent makes a mistake by picking the wrong Bluetooth speaker variant.

## 3.2 Turn-level Adjudicated Reinforcement Learning (TARL)

For a granular, turn-level assessment of each trajectory, we employ an LLM-based judge that evaluates every conversation turn (i.e., agent's reasoning, action, and environment feedback) against the ground-truth annotations. Prompt details of our judge are available in Appendix §E.

The judge's evaluation, visualized in Fig. 3, assigns one of three rewards: $-1$, $0$, or $1$, with the constraint that *at most one turn* can receive $-1$ per trajectory. A reward of $-1$ indicates a major deviation from expected behavior, typically occurring when the agent provides incorrect information after faulty reasoning (e.g., selecting the wrong item during an exchange request) or executes tool calls with erroneous arguments that cause irreversible database changes (e.g., canceling orders that should not be canceled). A reward of $0$ indicates minor issues that are later corrected or a by-product of major deviation. A turn receives 1 for correct execution without issues.

For GRPO, our final trajectory-level reward is a weighted combination of these turn-level scores ($r_i$) and the terminal (outcome) reward ($R(\boldsymbol{\tau})$) from our rule-based verifier. We scale the terminal reward $R(\boldsymbol{\tau})$ by $10\times$ to heavily prioritize successful task completion, and multiply the major deviation score ($-1$) by $5\times$ to penalize critical mistakes strongly, and scale all other turn scores by $1/T$ (where $T$ is the number of turns) to cap their contribution and prevent longer trajectories from being unfairly advantaged. Our reward design yields four distinct trajectory categories:

1. **Perfect trajectory** (15 points): 10 points for terminal success $+5$ points from turn-level rewards.
2. **Good trajectory** ($10-15$ points): 10 points for terminal success plus $0-5$ points from turn-level rewards, indicating some turns have minor issues.
3. **Good attempt trajectory** ($0-5$ points): 0 points for terminal failure but positive turn-level rewards, occurring when the judge finds no major errors despite rule-based verification failure (rare cases, often due to unclear or hallucinated user responses)
4. **Failed trajectory** ($-5$ to $0$ points): 0 points for terminal failure plus $-5$ points for one major error, with some positive reward from other turns.

Since PPO calculates advantages at the token level, we tested two reward granularities:

- **Per-Turn Assignment:** Applying each turn's reward specifically to the final token of that turn to provide more granular feedback, which will be propogated backwards by GAE (see Equation 4).
- **Trajectory-Level Assignment:** Calculating a single, normalized reward (using the same approach as GRPO) for the entire trajectory and applying it uniformly across all tokens.

By default, TARL for PPO uses trajectory-level assignment as it performs better (ablation available in §5). Beyond our core reward design, we also attempted several other strategies, including encouraging exploration with high-entropy token training (Wang et al., 2025) and utilizing turn-level verifiers to interrupt the reasoning process and force self-reflection. Though these strategies did not yield improvements, we discuss them in our analysis (§5) to provide insights for future research.

## 4 Experiments

### 4.1 Text-based Agent Training

**Training Data.** We train our text-based agents on approximately 3,000 synthetic tasks derived from APIGEN-MT (Prabhakar et al., 2025) trajectories. Each task provides: (1) a user instruction

| Agent Model | Response Metrics | | Performance Metrics | | | |
|---|---|---|---|---|---|---|
| | **#Wait** | **Len** | **pass^1** | **pass^2** | **pass^3** | **pass^4** |
| *Baseline Model* | | | | | | |
| Qwen3-8B | 14.6 | 228 | 42.6 | 30.4 | 25.2 | 21.7 |
| *Qwen3-8B + RL (GRPO Variants)* | | | | | | |
| GRPO | 11.7 | 204 | 51.3 | 37.4 | 30.4 | 27.0 |
| +TARL | 14.0 | 210 | 53.9 (+2.6) | 40.9 (+3.5) | 33.9 (+3.5) | 30.4 (+3.4) |
| +MATH + TARL | **15.8** | **236** | **57.4** (+6.1) | **42.6** (+5.2) | **36.5** (+6.1) | **33.9** (+6.9) |
| *Qwen3-8B + RL (PPO Variants)* | | | | | | |
| PPO | 8.4 | 162 | 48.7 | 36.5 | 31.3 | 26.1 |
| +MATH + TARL | 11.5 | 204 | 53.0 (+4.3) | 40.0 (+3.5) | 35.7 (+4.4) | **31.3** (+5.2) |

Table 2: Performance comparison of different training strategies on the $\tau$-BENCH RETAIL domain. We report average wait time (**#Wait**), average response length (**Len**), and pass^k metrics. Our proposed strategies (highlighted rows) consistently achieve the best performance

to guide the simulated user, and (2) the ground-truth tool calls the agent is expected to execute. Detailed construction process and examples are provided in Appendix §C. To ensure comprehensive coverage, our sandbox environment is also pre-populated with all seed data from $\tau$-BENCH. For our mixed-task training strategy, we incorporate math problems from the DeepScaleR dataset (Luo et al., 2025), filtering for problems with integer answers and alternating between RETAIL and math tasks during training. We will open-source all curated task instructions and ground-truth tool calls.

**Model.** We use Qwen3-8B (Yang et al., 2025) as the base model for our experiments. When using our proposed Turn-level Adjudicated Reinforcement Learning (TARL), we employ GPT-4.1 as the LLM judge to score each turn, following the mechanism described in §3.2. For full training hyperparameters, please refer to Appendix §D.

**Results** As shown in Table 2, our proposed Turn-level Adjudicated Reinforcement Learning (TARL) strategy, especially when augmented with mixed-task math training, consistently outperforms standard reinforcement learning baselines like GRPO and PPO[2]. Our optimal method (Math+TARL) achieves a 57.4% pass^1 score, representing a 6% relative improvement over GRPO and 15% over the base model. This result is competitive with capable closed-source models like GPT-4.1 (see Table 1), and the performance gains hold across different values of k, indicating enhanced reliability. Qualitatively, our method also produces models that engage in more frequent self-correction (higher #Wait tokens) and generate longer responses (**Len**), as detailed in Table 2. We provide further analysis of training statistics and alternative strategies in §5.

## 4.2 MULTIMODAL AGENT SETUP

**Environment and Simulation** To extend our framework to voice-driven interaction, we simulate realistic user speech by first generating textual user prompts and then converting them to audio using SeedTTS (Anastassiou et al., 2024), a high-quality text-to-speech model. This allows us to train agents on interleaved speech-text rollouts.

For evaluation, we assess the model in both text and speech modes. For the text mode, all settings are the same as text agents. For speech-mode evaluation, **we exclude the authentication step** from the RETAIL task in $\tau$-BENCH, as this step requires the agent to obtain user IDs in "name_number" format, which proves error-prone when processed through our TTS pipeline. Instead, we directly provide the agent with the user profile and continue the conversation.

**Model Selection and Baseline Performance** Our first step was to select a suitable base model capable of processing both speech and text. We evaluated several state-of-the-art foundational models, including Qwen2.5-Omni (Xu et al., 2025), Audio-Flamingo3 (Goel et al., 2025), and Audio-Reasoner (Xie et al., 2025). We found that none of these models demonstrated satisfactory tool-use capabilities out-of-the-box. While Audio-Flamingo3 and Audio-Reasoner struggled significantly, often hallu-

---

[2] TARL uses the trajectory-level assignment for PPO. Ablations on reward granularity are conducted in §5

| Training Configuration | | | Performance Metrics | | | |
|---|---|---|---|---|---|---|
| Eval Mode | Agent | Train Mode | pass^1 | pass^2 | pass^3 | pass^4 |
| *Baseline Models* | | | | | | |
| Text | Qwen2.5-Omni-7B | — | 7.8 | 7.8 | 7.8 | 7.8 |
| Speech | Qwen2.5-Omni-7B | — | 14.8 | 8.7 | 5.2 | 5.2 |
| *Qwen2.5-Omni-7B + RL* | | | | | | |
| Text | GRPO + Math | S & T | 31.3 | 20.9 | 12.2 | 12.2 |
| | GRPO + Math + TARL | S & T | **36.5** | **25.2** | **21.7** | **16.5** |
| Speech | GRPO + Math | S & T | 34.8 | 25.2 | 21.7 | 16.5 |
| | GRPO + Math + TARL | S & T | **37.4** | **26.1** | **22.6** | **20.9** |
| | GRPO + Math + TARL | T-only | 32.2 | 18.3 | 14.8 | 11.3 |

Table 3: Performance comparison across training and evaluation modalities on $\tau$-BENCH. Models are trained with speech-text (S-T) or text-only (T-only) rollouts and evaluated with text or speech-based user agent. Our proposed methods (highlighted rows) achieve the best performance.

cinating after one or two turns, Qwen2.5-Omni-7B achieved the best—though still poor—initial performance with a pass^1 rate of 7.8% (see Table 3). This highlights that multi-turn, interactive tool-use remains an under-explored capability for most speech-enabled foundation models.

**Curriculum Learning for Warming Up** Given the models' limited initial abilities, we adopted a curriculum learning strategy to warm-up the multimodal agent's tool-use abilities. Instead of supervised fine-tuning, we applied GRPO for 30 steps using a simplified set of training tasks. These tasks feature more detailed and specific user instructions to create a easier learning environment for skill acquisition (see Appendix §C). Qwen2.5-Omni showed rapid improvement on this simplified curriculum, demonstrating its ability to correctly use tools and engage in multi-turn conversations.

After the curriculum learning phase, we train the model on our normal training dataset and evaluate its performance across both text and speech modalities. During training, we employ a mixed-modality training strategy where the dataloader alternates between three types of data batches: (1) math problems, (2) text-only RETAIL task, and (3) RETAIL task with user response in speech. The first two batch types follow the same configuration used when post-training text agents. For the third data type, the model explores with interleaved speech-text rollouts where the speech contents are generated by the simulated user agent.

### 4.3 MULTIMODAL TRAINING RESULTS

The results in Table 3 validate the effectiveness of our proposed training strategy. Our final model, GRPO + MATH + TARL, consistently delivers superior performance across both text and speech evaluation settings, achieving a pass ^1 improvement of over 20% compared to the baseline. While the multimodal agent's performance currently lags behind its text-only counterparts, we anticipate this gap will narrow as foundational multimodal models continue to advance.

Crucially, an ablation study highlights the necessity of our mixed-modality training approach. When a model was fine-tuned exclusively on text and then evaluated in the speech-based setting, its performance degraded substantially (see final row of Table 3). This finding demonstrates that fine-tuning solely on textual data can erode a model's pre-trained speech understanding capabilities, underscoring the importance of using interleaved speech-text rollouts to develop effective voice agents.

## 5 ANALYSIS

### 5.1 REWARD GRANULARITY FOR PPO-BASED TRAINING

Given that PPO supports token-level rewards, we investigate how different reward granularities affect training performance. After obtaining turn-level evaluation from our judge, we experiment with two granularities as mentioned in §3.2: (1) *TARL (turn-level)*: assigning per-turn rewards at the fi-

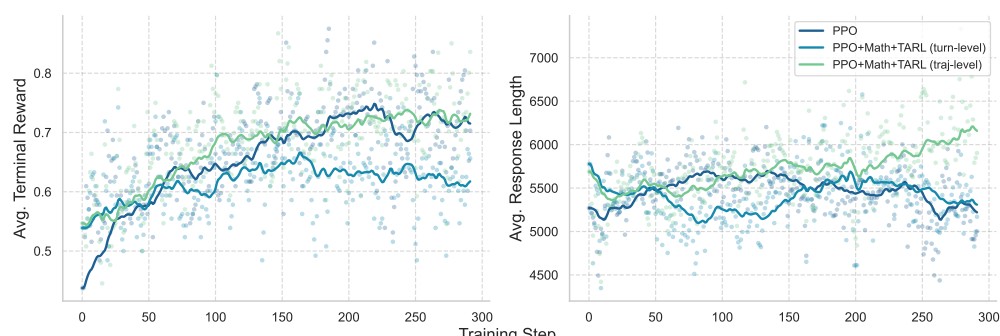

Figure 4: Training time average reward and response length comparison of PPO-based strategies. Trajectory-level assignment with turn-level eval (TARL traj-level) obtains the best performance.

nal token position of each turn and (2) *TARL (trajectory-level)*: computing a single trajectory-level reward (same as GRPO) and applying it uniformly across all token positions.

As illustrated in Fig. 4, the trajectory-level approach promotes more effective exploration and exhibits stable reward growth during training, ultimately achieving a 4.3% improvement in pass^1 performance compared to vanilla PPO training (see Table 2). In contrast, assigning rewards at turn-level granularity leads to degraded performance, with training rewards falling below even the vanilla PPO baseline. We hypothesize that assigning turn-level rewards at different positions complicates the credit assignment process and overly relies on the judge's accuracy. It is also sensitive to PPO hyperparameters that affect the discounting behavior of GAE. On the contrary, trajectory-level reward is much more robust as they are broadly dissected into four categories outlined in §3.2.

## 5.2 STRATEGIES FOR INCENTIVIZING EXPLORATION

**Data Distribution Modification: Mixed-Task Training**    We first examine the effectiveness of mixed-task training with mathematical problems. As shown in Fig. 5, GRPO+MATH demonstrates increased exploration activity during training, evidenced by longer average response lengths compared to the baseline. However, despite this enhanced exploration, test set performance remains comparable to the GRPO baseline, *suggesting that exploration alone is insufficient for improved generalization*. The combination of exploration strategies with better credit assignment proves crucial. GRPO+MATH+TARL, which incorporates both mixed-task training and turn-level rewards, exhibits the highest exploration levels (reflected in the longest average response lengths) and achieves substantially better performance on test set tasks (Table 2). Notably, all methods—GRPO, GRPO+MATH, and GRPO+MATH+TARL—converge to similar high reward levels during training (Fig. 3), indicating that the benefits of enhanced exploration and credit assignment primarily manifest in generalization to unseen RETAIL tasks rather than training performance improvements.

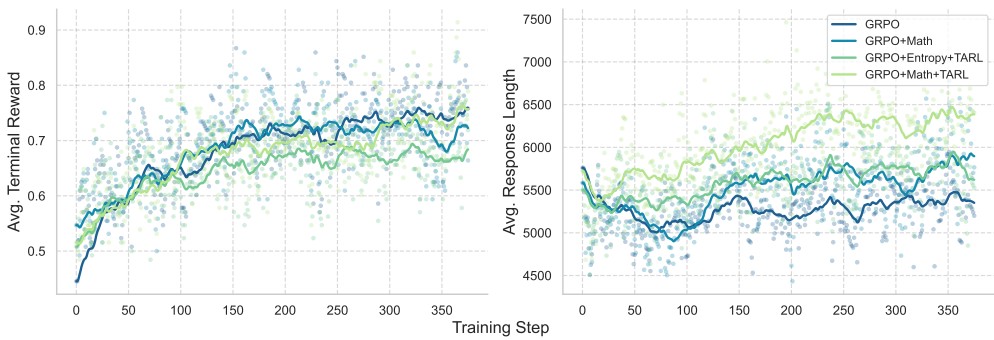

Figure 5: Training time average reward and response length comparison of different strategies. Mixed-task training with turn-level evaluation (GRPO+Math+TARL) achieves the best performance.

**Loss Function Adjustment: Entropy-based Modification**    We are also curious if loss function adjustment with entropy-based modification could help incentivize exploration. We follow the recent study (Wang et al., 2025) to restrict policy gradient updates to the top 20% highest-

entropy tokens. While this approach shows improved exploration compared to the baseline (see GRPO+Entropy+TARL in Fig. 5), it fails to enhance test-time performance and actually achieves lower training rewards than other strategies. We hypothesize that though entropy-based modification helps the model to explore, limiting updates to high-entropy positions could cause training instability, particularly problematic for our long-horizon sequential decision-making tasks.

In Appendix §F, we have additional analysis on the rollout intervention where we attempt to encourage exploration by forcing self-reflection when an erroneous tool-call is made. It turns out that editing the rollout context during training results in unstable updates that harm the performance.

**Key Takeaways**: Our mixed-task training strategy, when combined with a trajectory-level assessment that integrates both turn-level and terminal rewards, promotes more effective exploration and yields higher task completion rates. In contrast, more sophisticated interventions like complex reward shaping and elaborate training loss designs tend to destabilize the training process and ultimately degrade performance—a finding that echoes the "bitter lesson" (Sutton, 2019).

## 6 RELATED WORK

**Tool-Use Agent Benchmarks** Numerous evaluation benchmarks have been developed for tool-use tasks, including $\tau$-BENCH (Yao et al., 2024), $\tau^2$-BENCH (Barres et al., 2025), BFCL (Patil et al., 2025), AppWorld (Trivedi et al., 2024), ToolSandbox (Lu et al., 2025), UserBench (Qian et al., 2025), and Ace-Bench (Chen et al., 2025a). In our work, we adopt $\tau$-BENCH for training and evaluation as it supports realistic user-agent interactions, making it suitable for testing an end-to-end voice agent. However, $\tau$-BENCH has limitations, including its narrow scope of tasks (supporting only 2 domains) and limited control over user behavior. More recent benchmarks like UserBench have begun addressing these issues through preference-driven interactions, and we expect continued work in this direction to provide more controllable sandboxes for interactive tool-use tasks.

**Training Tool-Use Agents** Reinforcement learning (RL) algorithms have been developed and tested on a wide spectrum of problems. Foundational work demonstrated success in classic control tasks and games, such atari games (Mnih et al., 2015), and AlphaGo (Silver et al., 2016). More recently, RL has become a cornerstone for refining large language models (LLMs) beyond standard pre-training. Techniques like Reinforcement Learning from Human Feedback (RLHF) were critical in aligning models to follow user instructions and enhance safety (Ouyang et al., 2022). This paradigm has been extended to improve complex reasoning abilities, such as solving mathematical problems by rewarding correct final outcomes (Shao et al., 2024) or verifying intermediate reasoning steps with process reward modeling (Lightman et al., 2023). RL has also been applied to agentic tasks, such as WebShop (Yao et al., 2022; Zhou et al., 2024; Putta et al., 2024), AppWorld (Chen et al., 2025b), etc., with a simulated environment.

Addressing the credit assignment challenge in multi-turn interactions is difficult when using only final outcome-based rewards, despite their scaling potential (Shao et al., 2024; Zhang et al., 2025). Recent studies have shown that turn-level feedback offers a more effective solution for tool-use agents (Zhao et al., 2025; Zeng et al., 2025; Zhou et al., 2025). Building on insights from Process Reward Modeling (PRM) (Lightman et al., 2023; Ma et al., 2023; Zhang et al., 2025; Choudhury, 2025), we implement a turn-level reward system. Unlike previous approaches that rely on structured, rule-based evaluators (Zeng et al., 2025; Zhao et al., 2025), our method employs an LLM as a judge to provide more nuanced feedback (such as distinguishing between small and recoverable error versus major deviation) on an agent's performance at each turn.

## 7 CONCLUSION

We develop an interactive tool-use agent that communicates with simulated users and tool sandboxes to complete complex tasks. Through our carefully crafted environment, we enable the agent to perform online exploration and train it using reinforcement learning algorithms. We further enhance the learning process by incorporating mixed-task training to sustain exploration and employing turn-level evaluation to improve credit assignment in long-horizon tasks. Furthermore, we extend our framework to train multimodal voice agents, incorporating additional strategies such as curriculum learning and mixed-modality training to enhance agent performance across different modalities.

## ETHICS STATEMENT

Our work focuses on developing interactive tool-use agents, including multimodal voice agents capable of executing tasks based on spoken commands. While this technology holds promise for creating more natural and efficient human-computer interaction, it also introduces potential risks. Agents that can perform actions like modifying or canceling orders through API calls could be exploited for unauthorized or malicious purposes if not properly secured. Furthermore, the development of voice agents that interact via synthesized speech raises the possibility of misuse for deceptive applications, such as impersonation or social engineering. To mitigate these risks, we advocate for the implementation of robust safeguards, including strict access controls, user confirmation for critical actions, comprehensive audit trails for agent activities, and the use of techniques like audio watermarking to identify synthetic speech.

## REPRODUCIBILITY STATEMENT

Our research is conducted using publicly available datasets, including APIGEN-MT (Prabhakar et al., 2025) and DeepScaleR, in accordance with their respective licensing terms. The base models used in our experiments, such as the Qwen series (Xu et al., 2025; Yang et al., 2025) and various foundational models, are developed by third parties. Our user simulator and LLM-based judge leverage models like GPT-4.1 (OpenAI, 2024) and SeedTTS (Anastassiou et al., 2024). We promote transparency by providing the detailed judging prompt in Appendix §E. In the spirit of reproducibility and to encourage further research, we plan to open-source all synthetically generated task instructions and ground-truth tool calls created for this work. Our hyperparameters can be found in Appendix §D and we plan to open-source our codebase for reproducible experiments.

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

## A  SANDBOX ENVIRONMENT

Our sandbox environment, illustrated in Fig. 2, comprises three components, including (1) a backend application with a pre-configured database and API endpoints for MCP server communication; (2) a user simulator that leverages LLM capabilities to generate realistic user requests and responses; and (3) a rule-based verifier that evaluates interaction trajectories and provides binary rewards. Below, we detail each component's implementation and functionality.

**Backend Application.** We implement a SQLite database to store the comprehensive dataset from $\tau$-BENCH, encompassing various data tables such as Products, Orders, and Users. Rather than relying on static JSON files for seed data storage, we construct a proper relational database schema with well-defined table structures and database operations. This design choice enables our backend application to be easily extended and adapted for other tasks. We expose the available tools as RESTful API endpoints through application routers and register them as MCP tools, providing seamless integration for agent interactions.

**User Simulator.** Our user simulator employs GPT-4 (OpenAI, 2024) to role-play as human users, generating contextually appropriate requests and responses based on the task instructions from $\tau$-BENCH. We adopt the ReACT (Yao et al., 2023) reasoning framework using a consistent prompt with $\tau$-BENCH, which compels the user model to engage in structured thinking processes before formulating responses to agent queries. For speech-based user simulation, we use SeedTTS (Anastassiou et al., 2024) to convert the text responses to into natural speech.

**Rule-based Verifier.** We implement a rule-based verifier that systematically inspects successful write operations—specifically, tool calls that alter the database state, such as those involved in order modifications, exchanges, reservations, and cancellations. This verifier cross-references the arguments from the agent's tool calls with ground-truth annotations and outputs a binary reward: 1 for a complete match and 0 otherwise.

Notably, $\tau$-BENCH includes an additional verification step that checks for expected outputs in the agent's responses. However, we observe that this criterion is highly sensitive to variations in how user responses are phrased, so we exclude it from our reinforcement learning (RL) training and evaluation protocols. For the sake of consistency, though, we also report results incorporating this output check in Appendix §G.

## B  RL ALGORITHMS

In this section, we provide more detailed and formal description of the RL algorithms we adopted in our interactive tool-use scenario:

We formulate the interactive tool-use agent training as a Markov Decision Process (MDP). Given an autoregressive language model as the policy backbone, the state at any point in the interaction is simply the token sequence observed so far. The interaction follows an alternating pattern: when the agent is responding (calling tools with arguments), it takes actions by sampling the next token from the policy distribution $p_\theta$ and appending the token to the existing trajectory. When the agent stops talking, the environment generates feedback (through simulated user agent or tool execution results) and appends a sequence of tokens (denoting user response or tool execution result) to the existing trajectory. More formally, let $\boldsymbol{x}^i = (x_1^i, x_2^i, \ldots)$ denote the $i$-th agent token sequence and $\boldsymbol{e}^i = (e_1^i, e_2^i, \ldots)$ denote the $i$-th environment token sequence. When the environment response is from tool execution or text-based user simulation, $\boldsymbol{e}^i$ is a sequence of text tokens. When we use speech-based user simulation, $\boldsymbol{e}^i$ is a sequence of speech tokens (or their placeholder tokens). The complete trajectory is an interleaved sequence: $\boldsymbol{\tau} = (\boldsymbol{x}^1, \boldsymbol{e}^1, \boldsymbol{x}^2, \boldsymbol{e}^2, \ldots, \boldsymbol{x}^T, \boldsymbol{e}^T)$. Here $T$ is the total number of interaction steps, reached when user agent replied special token ##STOP## or when the maximum number of interaction steps is reached. In our case, $T \in [1, 30]$ as we set a maximum of 30 interaction steps. Our objective is to maximize the expected reward over complete trajectories:

$$J(\theta) = \mathbb{E}_{\boldsymbol{\tau} \sim p_\theta}[R(\boldsymbol{\tau})] \tag{1}$$

where $R(\boldsymbol{\tau})$ is a trajectory-level reward function that evaluates the quality of the generated trajectory using the rule-based verifier described in §2.1. To optimize this objective, we experiment with the following widely-used RL algorithms:

**PPO (Proximal Policy Optimization)**: We begin with PPO (Schulman et al., 2017), which constrains policy updates to prevent large deviations from the current policy through a clipping mechanism. For brevity, we denote the conversation history up to token $x_j^i$ as $\boldsymbol{h}_j^i = [\boldsymbol{c}; \boldsymbol{x}^1, \boldsymbol{e}^1, \ldots, \boldsymbol{x}^{i-1}, \boldsymbol{e}^{i-1}, x_1^i, \ldots, x_{j-1}^i]$. PPO operates at the token level using policy gradient ratios. Given a current policy $\theta_{\text{old}}$ and a new policy $\theta$, the probability ratio for each token is:

$$r_j^i(\theta) = \frac{p_\theta(x_j^i | \boldsymbol{h}_j^i)}{p_{\theta_{\text{old}}}(x_j^i | \boldsymbol{h}_j^i)} \tag{2}$$

The PPO objective function applies clipping to this ratio (we omit the KL divergence term here):

$$L^{\text{PPO}}(\theta) = \mathbb{E}_{\boldsymbol{\tau} \sim p_{\theta_{\text{old}}}} \left[ \frac{1}{\sum_{i=1}^{T} |x^i|} \sum_{i=1}^{T} \sum_{j=1}^{|\boldsymbol{x}^i|} \min \left( r_j^i(\theta) A_j^i, \text{clip}(r_j^i(\theta), 1-\epsilon, 1+\epsilon) A_j^i \right) \right] \tag{3}$$

where $A_j^i$ is the advantage function and $\epsilon$ is the clipping parameter. PPO uses the Generalized Advantage Estimate (GAE):

$$A_j^i = \sum_{l=0}^{\infty} (\gamma\lambda)^l \delta_{j+l}^i, \quad \text{where} \quad \delta_j^i = R_{j+1}^i + \gamma V(s_{j+1}^i) - V(s_j^i) \tag{4}$$

This formula relies on several key terms. The calculation is driven by the Temporal Difference (TD) error ($\delta_j^i$), which measures the one-step prediction error of the value function. The TD error itself is found using the immediate reward ($R_{j+1}^i$) and the value function's estimate for the current and next states. This calculation is weighted by two parameters: the discount factor ($\gamma$), which determines how much future rewards are valued, and the GAE parameter ($\lambda$), which balances the trade-off between bias and variance in the final advantage estimate[3]. Since we use a trajectory-level verifiable reward, we simply have $R_{j+1}^i = R(\boldsymbol{\tau})$, i.e., *the token-level reward is the same across all positions*. Note that throughout our RL training, we compute loss only over agent sampled tokens $\boldsymbol{x}$ and mask the loss over all environment tokens $\boldsymbol{e}$ to avoid unstable updates.

**GRPO (Group Relative Policy Optimization)**: GRPO (Shao et al., 2024) also shares the same clipping mechanism but introduces a group-based relative policy optimization approach that normalizes rewards within each group to improve training stability. Given a group of $G$ trajectories $\{\boldsymbol{\tau}_1, \boldsymbol{\tau}_2, \ldots, \boldsymbol{\tau}_G\}$, GRPO computes the mean and standard deviation of rewards:

$$\mu_R = \frac{1}{G} \sum_{n=1}^{G} R(\boldsymbol{\tau}_n), \quad \sigma_R = \sqrt{\frac{1}{G} \sum_{n=1}^{G} (R(\boldsymbol{\tau}_n) - \mu_R)^2} \tag{5}$$

The advantage for trajectory $n$ is then computed as $A_n = (R(\boldsymbol{\tau}_n) - \mu_R)/\sigma_R$. The GRPO objective function is:

$$L^{\text{GRPO}}(\theta) = \mathbb{E}_{\{\boldsymbol{\tau}_n\}_{n=1}^{G} \sim \mathbb{P}_{\theta_{\text{old}}}} \left[ \frac{1}{\sum_{i=1}^{T_n} |x^i|} \sum_{i=1}^{T_n} \sum_{j=1}^{|\boldsymbol{x}^i|} \min \left( r_j^{i,n}(\theta) A_n, \text{clip}(r_j^{i,n}(\theta), 1-\epsilon, 1+\epsilon) A_n \right) \right] \tag{6}$$

where $r_j^{i,n}(\theta) = \frac{p_\theta(x_j^{i,n} | \boldsymbol{h}_j^{i,n})}{p_{\theta_{\text{old}}}(x_j^{i,n} | \boldsymbol{h}_j^{i,n})}$ is the probability ratio for token $j$ in turn $i$ of trajectory $n$. This normalization approach helps stabilize training by reducing reward scale variations across different batches and tasks.

**RLOO (REINFORCE Leave-One-Out)**: RLOO (Ahmadian et al., 2024) is similar to GRPO but uses a different baseline computation. Given a group of $G$ trajectories $\{\boldsymbol{\tau}_1, \boldsymbol{\tau}_2, \ldots, \boldsymbol{\tau}_G\}$, the baseline for each trajectory $\boldsymbol{\tau}_n$ is computed using all other trajectories in the group:

$$b_n = \frac{1}{G-1} \sum_{j \neq n} R(\boldsymbol{\tau}_j) \tag{7}$$

Then the advantage function is computed as $A_n = R(\boldsymbol{\tau}_n) - b_n$ for the $n$-th trajectory. This leave-one-out approach ensures that the baseline is unbiased while significantly reducing the variance of gradient estimates compared to standard REINFORCE.

---

[3] In practice, we set $\gamma = 1, \lambda = 1$ given our long-horizon trajectories

## C   TRAINING DATA

| User Specification | Ground-Truth Tool-Calls |
|---|---|
| Your email is noah.brown7922@example.com. You are a customer who recently received an order and want to exchange two items: the green small polyester laptop-compartment backpack for a navy large polyester laptop-compartment backpack, and the black wired laser gaming mouse for a black wireless optical gaming mouse ... Describe the items you want to exchange and the new options you want, and confirm the use of your original payment method for any price difference only after the agent identifies it. Respond to the AI agent to complete your exchange request. | `[...(other tool-calls),`
`{`
`  "name": "exchange_delivered_order",`
`  "arguments": {`
`    "order_id": "#W7678072",`
`    "item_ids": [`
`        "3557711149",`
`        "2193628750"`
`    ],`
`    "new_item_ids": [`
`        "8084436579",`
`        "8214883393"`
`    ],`
`    "payment_method_id": "paypal_xxx"}`
`}]` |

Table 4: Example of synthetic training data showing user specification and corresponding ground-truth tool calls for an item exchange scenario.

We generate synthetic training tasks by leveraging conversation trajectories from APIGEN-MT (Prabhakar et al., 2025) and using large language models to synthesize corresponding user specifications (instructions) with the prompt below. An illustrative example of this process is presented in Table 4. In practice, we employ GPT-4.1 to extract user specifications from the conversation trajectories provided by APIGEN-MT. The prompt to craft such specifications is shown below:

---

**User Specification Synthesis Prompt Template**

**Role & Objective:** You are an expert in analyzing human-AI conversation trajectories. Your task is to infer the *core instruction* or *task* that the human user was given, which led to their interaction with the AI.

**Analysis Framework:** Analyze the provided conversation transcript, paying close attention to:
 1. *Initial Request:* The human's opening statement or question
 2. *Response Patterns:* How the human responds to AI queries
 3. *AI Actions:* Function calls, observations, and AI responses that reflect user intent
 4. *Conversation Flow:* Overall progression and resolution

**Output Requirements:** Based on your analysis, formulate a concise, direct, and clear instruction that, if given to a human, would result in the conversation you observe. The instruction must capture:
 – User's role/identity
 – User's objective/task
 – Reason/context for the task
 – Key constraints or requirements

**Output Format Template:** *"Your user id is [user_id or email if available in conversation]. You are [User's Role/Identity] and you are trying to [User's Objective/Task] because [Reason/Context]. You need to provide [Key Information Required] and respond to the AI's prompts to achieve your goal."*

**Input:** *[CONVERSATION_TRANSCRIPT]*

**Expected Output:** *[SYNTHESIZED_USER_INSTRUCTION]*

---

While the template above enables us to synthesize high-quality user instructions for each task, these instructions tend to conform to a similar format due to our structured "output format template". To introduce greater diversity and increase the exploration challenge for the model, we rewrite the synthesized instructions using the following template. We rewrite the instruction to be more challenging for the agent to complete, while ensuring that the tasks remain solvable. The re-writing prompt is shown below:

**User Instruction Rewriting Prompt Template**

You will be provided with a user-agent conversation trajectory and a user instruction.
You job is to re-write the user instruction following the steps below:

---

1.  You should first read through the conversation between user and agent, understanding
the user's intention and from the AI agent's reply, you will have detailed information
such as the user's information and order details.  Pay special attention to the function
calls and the arguments in each function call.

2.  You should then read through current user instruction, the insurction already
provides necessary and detailed information to the user to complete the conversation
with the agent.

3.  Now your job is to re-write the user instruction so that the user withhold certain
information from the agent, but the task should still be possible to complete even
without those withheld information, because such information might be retrieved from
other function calls.

For example, get_user_details will show not only user information but also payment_methods
and the user's current order ids.  Therefore, even if the user forgets order ID, it
can be retrieved and confirmed by the agent.  Similarly, get_order_details will return
order_id, user_id, user_address, order items, as well payment_history, payment status, and
fulfillments.  These information can then be used to help process the order even if the
user forgets some details about their order or address.

Here are some ways to re-write the instruction and make it harder for the agent:
- You can ask the user not to provide order details (say that you do not remember it)
but ask the agent to derive it from its user profile
- You can ask the user not to provide payment method (say that you do not remember it)
but only to confirm after agent replies with options
- You can ask the user not to provide details they want (say that you do not remember
it) but only expose them after such items/products are provided by the agent as options
- Be creative and think of any other ways to make the instruction harder (but please
make sure that the task is possible to complete)

Do not make the task too hard, only randomly apply one or two withholding strategies
above in your re-write process.  Also, please ensure that the user instruction contains
necessary information (order ids, payment methods, etc.,) even if the user does not
provide it explicitly.  The agent will always authenticate the user's identity first so
please make sure that the user information is provided in the instruction:
    - User information could be user_id, user name + zipcode or user email.
    - You can modify the instruction so that user withholh some of their user
information, but at least one of these information should be possible to be obtained
by the agent (e.g., user can forget user_id and zipcode but provide email for
authentication)

4.  When you re-write the instruction to be more challenging for the agent, please
make sure the original information necessary for the user to complete the task is still
provided to user.  For example, even if you ask the user to withhold their user_id,
address, payment_method, order details, etc., you should still provide these information
in the instruction (so that user still knows about them even if they will not provide
information explicitly to the agent).

---

**Output Requirements:**

<think>
You should conduct an evaluation of the user instruction and think about how to re-write
the instruction based on my rules above inside this block
</think>

Then you need to output a json object with the following fields:
{
    "rewrite_instruction":  <your re-written instruction>
}

Note that in section §4.3, we mentioned that for multimodal warm-up training, we use a simpler version of the training tasks. To create this simple version, we use the following prompt template:

---

**Simplification Prompt Template for Multimodal LLM Warm-up Training**

```
You will be provided with a user-agent conversation trajectory and a user instruction.
You job is to re-write the user instruction following the steps below:

1.  You should first read through the conversation between user and agent, understanding
the user's intention and from the AI agent's reply, you will have detailed information
such as the user's information and order details.
2.  You should then read through current user instruction, and check if it provides
enough information for the user to complete the conversation with the agent.  Pay
special attention to the conversation where the agent is asking for user's information
or confirmation about choices.
3.  Try to re-write the user instruction to be more detailed.  Pay special attention
to the arguments in each function calling.  User information, order number and details,
payment information should all be included in the instruction if available.

Output Requirements:
<think>
You should conduct an evaluation of the user instruction and think about how to re-write
the instruction based on my rules above inside this block
</think>

Then you need to output a json object with the following fields:
{
    "rewrite_instruction":  <your re-written instruction>
}
```

---

# D    TRAINING DETAILS

We adopt verl[4] and RL Factory [5] as our framework to support RL training with multi-turn conversation. We train the model with a batch size of 128 (e.g., 32 distinct tasks with 4 rollouts per task). We train the agent model until convergence (performance normally plateaus after 200-300 steps) with hyperparameters shown in Table 5.

| Hyperparameter | Value | Hyperparameter | Value |
|---|---|---|---|
| grad_clip | 1.0 | max_prompt_length | 4096 |
| clip_ratio | 0.2 | max_response_length | 1024 |
| ppo_epochs | 1 | kl_coef | 0.001 |
| num_rollout | 4 | kl_loss_coef | 0.003 |
| top_p | 0.95 | actor_lr | $1 \times 10^{-6}$ |
| temperature | 0.7 | critic_lr | $1 \times 10^{-5}$ |
| max_turns | 30 | | |

Table 5: Hyperparameter settings used in our experiments.

# E    LLM JUDGE SETUP FOR TURN-LEVEL EVALUATION

To assess the multi-turn trajectory, the LLM-based Judge will receive the complete trajectory and ground-truth tool-call annotations to output a score for each turn. Below is the prompt that we use to provide turn-level rewards:

---

[4]https://github.com/volcengine/verl
[5]https://github.com/Simple-Efficient/RL-Factory

**Role: Task Execution Evaluation Judge**

Your core responsibility is to **thoroughly and precisely evaluate multi-turn conversations** between a user and an agent. You must carefully read each conversation to pinpoint where the agent's decisions lead to deviations from the ground-truth function-call trajectories.

**Information Provided for Your Evaluation**

You will be given four key pieces of information to guide your assessment:

1. Policy: This document outlines the **strict rules the agent must adhere to** when making tool calls. If an agent's action violates this policy, you must immediately halt its current action and instruct it to reconsider and correct its approach.

2. Task Instruction: This is the **specific instruction provided to the user.** The user's requests and responses should always align with this instruction. **The agent does not have access to this instruction.**

3. Ground-Truth Function Call Trajectories: This serves as the **definitive standard** for assessing the accuracy of the agent's tool calls.

   - The agent **doesn't need to follow the exact order** of this trajectory.
   - It's acceptable for the agent to call information-gathering functions (e.g., get_order_details) multiple times, but the agent's write operation (modifying, exchanging, returning, or canceling orders) needs to match exactly with the ground-truth function calls.

4. Conversation Trajectories: This provides the **detailed record of the multi-turn conversation** between the user and the agent. You will use these conversations to identify executed tools and evaluate the correctness of the agent's processing of results. Each agent's reasoning and action within a turn will be preceded by a label like [Turn N].

**Evaluation Process**

Deviations from the ground truth typically arise due to:

• The agent failing to gather sufficient or correct information, either through function calls or by asking the user.

• Incorrect reasoning or understanding by the agent based on the results of tool execution.

• The agent not following policy, resulting in wrong execution of tools.

Pay **exceptionally close attention to operations involving modifying, exchanging, returning, or canceling orders.** The agent's calling for these function should match exactly with the ground-truth. These are critical evaluation points and frequent sources of error. Three kinds of error are possible with write operation:

(1) The agent might call the function with wrong arguments that do not match with ground-truth.

(2) The agent calls unnecessary write operation that should never be called.

(3) The agent did not call the write operation which is listed in the ground-truth.

If any of the three cases above occurs, you need to carefully read the conversation and identify the turns where the agent deviates from the ground-truth.

For each turn in the conversation (identified by the [Turn N] tag), you should evaluate whether agent's reasoning and action in that turn is the primary cause of a deviation from the ground-truth function call. Assign a score for each turn. You have three kinds of score to assign:

• If the turn is correct, assign a score of 1.

• If the turn is the primary reason for a deviation, assign a score of -1. This can only be assigned to at most one of the conversation turns if deviation is found.

• If the turn has issue (e.g., not following the policy or function call formats), assign a score of 0.

**Your Response Format**

You **must** first conduct your evaluation process within a <think></think> block.

After completing your thinking process, you **must** output only a single JSON object. No other text, commentary, or explanation should be included outside of the JSON block. The JSON object **must** adhere strictly to the following format, including all turn's scores from score_0 up to score_n (where n is the total number of turns).

```
{
    "score_0":  <turn 0's score>,
    "score_1":  <turn 1's score>,
    // ...  (include all turns up to 'n')
    "score_n":  <turn n's score>
}
```

# F   FORCING EXPLORATION WITH ROLLOUT INTERVENTION

Inspired by self-refinement techniques (Weng et al., 2023; Madaan et al., 2023), we also explored whether real-time intervention from a verifier could improve the agent's exploration through forcing self-reflections. We deployed an LLM-based verifier to continuously monitor the agent's action, as visualized in Fig. 6.

By comparing the agent's actions to ground-truth tool calls, the verifier can identify suboptimal reasoning as it happens. When a mistake is detected, it triggers a self-correction mechanism by interrupting the generation and adding a corrective prompt to the reasoning trace: "Wait, my previous reasoning might be wrong, let me try again." This prompts the model to find a better approach, with a limit of two interventions per reasoning

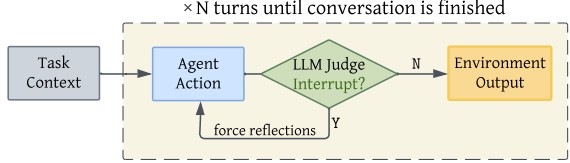

Figure 6: Rollout intervention with an LLM-based judge. When major deviations from expected ground-truth tool calls are detected, the judge will force the agent to think and act again.

step to avoid infinite loops. This real-time guidance is a more dynamic version of our turn-level reward system, which only provides feedback after a task is complete. However, this strategy ultimately backfired, destabilizing training without improving performance. As shown in Fig. 7, the rapid decrease in entropy loss, paired with a significant rise in KL divergence, suggests the model began to overfit the unusual data patterns created by the interruptions. We attribute this failure to the disruption of the model's natural thought process, which led to confusion and worse results than even the standard baseline.

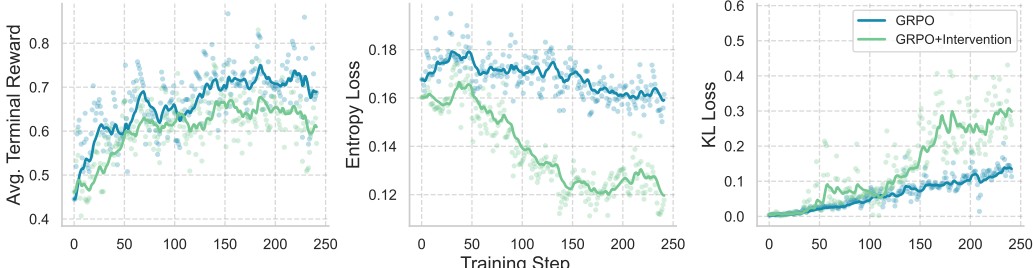

Figure 7: Training time average reward, entropy loss, and KL loss comparison between GRPO and GRPO+Intervention.

## G  MORE EXPERIMENTAL RESULTS

| Agent Model | Retail | | | | Airline | | | |
|---|---|---|---|---|---|---|---|---|
| | pass ^1 | pass ^2 | pass ^3 | pass ^4 | pass ^1 | pass ^2 | pass ^3 | pass ^4 |
| *Baseline Models* | | | | | | | | |
| **GPT-4.1** | 58.3 | 53.0 | 49.6 | 46.1 | 48 | 34 | 26 | 24 |
| **xLAM-2-8B** | 41.7 | 30.4 | 25.2 | 22.6 | 32 | 24 | 18 | 16 |
| **Qwen3-8B** | 40.0 | 27.8 | 22.6 | 18.3 | 30 | 20 | 18 | 18 |
| *Qwen3-8B + RL* | | | | | | | | |
| **GRPO** (n=4) | 47.0 | 35.7 | 27.8 | 24.3 | 28 | 12 | 6 | 2 |
| **RLOO** (n=4) | 44.3 | 29.6 | 26.1 | 21.7 | 34 | 18 | 14 | 10 |
| **PPO** (n=4) | 47.0 | 33.9 | 28.7 | 22.6 | 30 | 20 | 12 | 10 |
| **PPO** (n=1) | 47.0 | 26.1 | 18.3 | 15.7 | 30 | 16 | 8 | 6 |

Table 6: Results of tool-use agents trained with different RL algorithms on $\tau$-BENCH with output check.

| Agent Model | Avg. Wait | Resp. Len | pass^1 | pass^2 | pass^3 | pass^4 |
|---|---|---|---|---|---|---|
| **Qwen3-8B** | 14.6 | 228 | 40.0 | 27.8 | 22.6 | 18.3 |
| **GRPO** | 11.7 | 204 | 47.0 | 35.7 | 27.8 | 24.3 |
| **+Turn-Level Reward** | 14.0 | 210 | 52.2 | 39.1 | 29.6 | 26.1 |
| **+MATH + Turn-Level Reward** | 15.8 | 236 | **53.9** | **40.0** | **34.8** | **30.4** |
| **PPO** | 8.4 | 162 | 47.0 | 33.9 | 28.7 | 22.6 |
| **+MATH + Turn-Level Reward** | 11.5 | 204 | 52.2 | 39.1 | **34.8** | **30.4** |

Table 7: Performance comparison of different training strategies on the $\tau$-BENCH RETAIL domain (with output check). We report average wait time, response length, and pass^k metrics. Best results are highlighted in **bold**.

## H  THE USE OF LARGE LANGUAGE MODELS

We also acknowledge the use of AI assistants (e.g., GitHub Copilot, ChatGPT) to aid in the research process. These tools were utilized for tasks such as code implementation, debugging, and refining the manuscript. All core conceptual contributions, experimental design, and final analyses were conducted and validated by the authors to ensure scientific rigor and originality, in adherence with academic integrity standards.

