# OpenReview forum: "Process-Supervised Reinforcement Learning for Interactive Multimodal Tool-Use Agents"
_ICLR.cc/2026/Conference — Submitted to ICLR 2026_

### Official Review · Reviewer_y6Kx · 2025-10-25

**Soundness:** 2
**Presentation:** 2
**Contribution:** 4
**Rating:** 4
**Confidence:** 4

**Summary:**

This paper provides several interesting contributions to the community, namely: (1) a sandbox for text and speech training with RL and (2) ablations showing that training with *speech and text* with RL is more effective on both domains than training with text alone, (3) a method using trajectory level rewards designed by an LLM-as-a-judge annotating turn-level rewards that improves performance significantly on tao-bench, and (4) results demonstrating that mixed-task training and curriculum learning are an effective strategy for RL.

**Strengths:**

1. Introduce a sandbox environment combining speech and text in the tao bench environment.
2. Perform multimodal training on speech and text and find that speech and text training using RL is a powerful tool for improving performance on both the text alone set up and the multimodal configuration of the environment
3. The paper presents a mixed-task training curriculum wherein the authors mix together math problems with tao agent and multi-modal rollouts. The paper shows that including these math tasks significantly improves performance, an interesting finding for the community.
4. The paper provides a detailed analysis of using a trajectory level reward from turn level reward metrics and demonstrates that using this can yield substantial improvements on tao bench.
5. Ablate against other entropy based methods for turn level credit assignment and find them to not substantially improve performance on this domain.
6. Formulate an LLM-as-a-judge reward assignment paradigm, though the details of how the LLM-as-a-judge is implemented are lacking

**Weaknesses:**

1. The abstract, introduction & title of the algorithm all seem to indicate that turn-level (e.g. token level) reward assignment is critical to algorithm success. However, the most successful variation of this algorithm involves summing up all of the token level rewards and placing it at the end of the trajectory. I find it hard to believe that if we are summing up the reward and placing it at the end of the trajectory we are receiving the fine-grained, turn-level credit assignment claimed in the third bullet point of the introduction
2. The second critical component of this algorithm is the LLM-as-a-judge, but there is limited details on how we verify that the judgements made by the LLM-as-a-judge are truly useful other than the downstream RL performance
3. It seems that the LLM-as-a-judge requires a ground truth trajectory something that could be hard to find in many other domains (such as coding where there are many correct solutions)

**Questions:**

1. How is the confidence paradox addressed by the trajectory level rewards proposed in this paper? Do you observe any qualitative or quantitative results indicating
2. How did you verify that the LLM-as-a-judge was providing reliable answers on the turn level?
3. From the findings of this paper, do you believe that turn-level rewards are viable future research direction and we simply need to engineer these rewards to be more useful and the credit assignment algorithm to be better or should practictioners and researchers focus on more useful trajectory-level rewards and mixed task training?

This reviewer would be willing to give a higher score if the authors reframe the paper away from turn-level rewards, or some suitable explanation from the authors on why they chose to frame the paper in this way.

---

> ### Author Response · Authors · 2025-11-19
> **Author Response to Reviewer y6Kx**
>
> Thank you for the time and effort you've invested in reviewing our paper. We are grateful for your recognition of our approach's novelty and the improvements it offers. Below are our responses to your comments:
>
> > I find it hard to believe that if we are summing up the reward and placing it at the end of the trajectory we are receiving the fine-grained, turn-level credit assignment claimed in the third bullet point of the introduction. This reviewer would be willing to give a higher score if the authors reframe the paper away from turn-level rewards, or some suitable explanation from the authors on why they chose to frame the paper in this way.
>
> We thank the reviewer for raising this important point and we would like to provide additional clarification.
>
> First, **reward signal and reward placement are distinct notions**. Although we place the final reward at the end of the trajectory, the signal itself is computed at turn-level resolution. Our scoring scheme produces a weighted sum over these turn-level scores, grouping trajectories into four categories (Section 3.2). As a result, the agent still benefits from fine-grained credit assignment even though the scalar reward is attached to the final token.
>
> Second, modern PPO/GRPO-style post-training pipelines routinely attach the reward only at the final step, using the value function (PPO) or group-based advantage (GRPO) to propagate credit across the trajectory. This design has become standard because long-horizon LLM-agent training is substantially more stable when the reward is aggregated, whereas injecting sparse intermediate rewards often introduces instability.
>
> Third, our experiments reinforce this point: while turn-level signals are essential for long-horizon tasks, explicitly injecting per-step rewards performs worse than aggregating them at the end (Section 5.1). GRPO, in particular, benefits from treating a trajectory as a coherent unit. When supplied with our turn-level reward signal, its relative-advantage mechanism not only distinguishes good and bad samples, but also good vs. less good and bad vs. less bad, aligning with principles from preference learning and contrastive objectives. This explains why **trajectory-level placement delivers more stable training while still achieving higher performance** through turn-level signal.
>
>
> We hope this clarification addresses the reviewer’s concern, and we would be happy to revise the text further if needed.
>
>
> > There is limited details on how we verify that the judgements made by the LLM-as-a-judge are truly useful. How did you verify that the LLM-as-a-judge was providing reliable answers on the turn level?
>
> Thank you for raising this concern. We have added a human study evaluating the reliability of our LLM-Judge, and we kindly refer you to our general response to all reviewers (part 1) for full details. In summary, the LLM-Judge demonstrates strong consistency with human evaluation across tasks, exhibiting high correlation with human scores and highly accurate detection of deviation turns.
>
> > It seems that the LLM-as-a-judge requires a ground truth trajectory something that could be hard to find in many other domains (such as coding where there are many correct solutions)
>
> Thank you for raising this question. We would first like to clarify that our approach **does not require a ground-truth trajectory**. Instead, we rely only on the ground-truth tool calls (which correspond to one or a few essential actions that must appear somewhere in the trajectory) together with the sampled trajectory for the LLM-Judge to evaluate each turn.
>
> Similar to coding or other multi-step tasks, there are often multiple valid solution paths. Our design is fully compatible with this setting as long as some form of intermediate signal can be extracted. In our case, the LLM-Judge identifies major deviation turns; for coding or computer-use agents, intermediate code-execution results or environment-rendering feedback can be analyzed in the same way to yield turn-level reward signals. Recent works [1, 2, 3] already adopt similar strategies. Thus, the high-level approach naturally extends to other multi-turn tasks where intermediate signals are available.
>
> ---
>
> [1] Chang et al., 2025. THOR: Tool-Integrated Hierarchical Optimization via RL for Mathematical Reasoning
>
> [2] Dai et al., 2025. Process Supervision-Guided Policy Optimization for Code Generation
>
> [3] Tang et al., 2025. SEA: Self-Evolution Agent with Step-wise Reward for Computer Use

---

> > ### Author Response · Authors · 2025-11-19
> > **Author Response to Reviewer y6Kx**
> >
> > > How is the confidence paradox addressed by the trajectory level rewards proposed in this paper? Do you observe any qualitative or quantitative results indicating
> >
> > Thank you for the question. It appears the sentence was cut off, so we kindly ask for clarification on the full intended question.
> >
> > Regarding the confidence paradox, we introduce it as a motivation for improving exploration during post-training—specifically, to avoid premature entropy collapse where the model becomes overconfident too early. This issue is primarily mitigated through our mixed-task training with Math problems (Section 5.2), which we found far more effective than alternative attempts such as entropy-regularized training.
> >
> > That said, the turn-level signal provided by TARL also contributes to countering the confidence paradox. By supplying more informative feedback at the level of individual turns, TARL encourages higher-quality exploration throughout the trajectory. As shown in Figure 5, GRPO+MATH already produces longer and more exploratory trajectories than vanilla GRPO, but GRPO+MATH+TARL leads to even longer trajectories and better downstream performance. This suggests that **both enhanced exploration mechanisms and richer reward schemas are important** for addressing the confidence paradox we observed in our vanilla post-training setup.
> >
> >
> >
> > > Do you believe that turn-level rewards are viable future research direction…or should practictioners and researchers focus on more useful trajectory-level rewards and mixed task training?
> >
> > Thank you for the question. We view the components mentioned here—turn-level rewards, trajectory-level rewards, and mixed-task training—as complementary rather than mutually exclusive. In fact, our results suggest that these elements work best together to improve post-training performance.
> >
> > As discussed in our general response, we believe that as RL environments for LLM agents continue to scale and diversify, post-training pipelines will naturally evolve into multi-stage systems spanning heterogeneous task types, each requiring its own tailored reward schema.
> >
> > Some long-horizon tasks (such as ours) benefit from turn-level evaluation via an LLM-as-Judge; interactive tool-use agents, multi-turn customer-support agents, and computer-use agents (e.g., WebArena) require fine-grained judgments of intermediate steps. Other tasks are better suited to continuous reward vectors from learned reward models—for example, dialogue quality scoring, summarization preference models, or instruction-following reward models used in current RLHF pipelines, where a scalar or vector reward captures nuanced attributes like helpfulness, factuality, or style. Some tasks can rely on simple rule-based metrics, such as programming/code-generation, where unit-test pass rates provide a clear binary signal; or planning tasks with explicit constraints (e.g., Sudoku, path planning), where correctness can be checked without a learned model.
> >
> > In short, there is no single reward setup that generalizes across all tasks. Within an “environment zoo,” we believe **each environment should leverage the evaluation mechanism that best aligns with its structure and affordances**.

---

> > > ### Comment · Reviewer_y6Kx · 2025-11-26
> > > **Thank you**
> > >
> > > Thank you to the authors for your detailed rebuttal. This resolves many of my concerns. I've reflected on your response and my main concern and realized that my main issue revolves around this phrase in lines 194-197
> > >
> > > "Furthermore, the vanilla use of trajectory-level rewards could be problematic for multi-turn conversations—in our case, with contexts up to 32,768 tokens—as it creates sparse reward signals that lead to suboptimal credit assignment when the model performs multiple actions per trajectory"
> > >
> > > Your response highlights that trajectory-level/sparse rewards are common practice and I agree with this assertion, but this phrase in the paper seems to indicate that reward sparsity and trajectory reward assignment are an essential problem to be fixed. In order for me to raise my score, I would like you to rephrase this around the ideas of process rewards or learning better from token-level rewards, even if they are assigned at the end. Overall I think this will lead to a better reading experience for future readers so they don't feel as if your paper and results are contradicting one another. For example, something like this from your rebuttal would serve as good starting point for a rewrite:
> > >
> > > ```Trajectory-level placement delivers more stable training while still achieving higher performance through turn-level signal```
> > >
> > > I would be more than happy to raise my score if we can come to an agreement on this.

---

> > > > ### Author Response · Authors · 2025-11-26
> > > > **Author Response to Reviewer y6Kx**
> > > >
> > > > Thank you very much for the follow-up and for pointing out the source of confusion. We completely agree with your assessment that the phrasing in lines 194–197 unintentionally suggests a contradiction. To avoid the misleading implication, we will adopt language similar to what you suggested. A revised version of the paragraph will read along the following lines:
> > > >
> > > > > “While naive use of trajectory-level rewards can lead to sparse feedback in long-horizon conversations, our approach avoids this issue by deriving the reward from turn-level evaluations, then aggregating it at the end. This maintains the stability benefits of trajectory-level placement while still enabling fine-grained credit assignment. In practice, we find that trajectory-level placement delivers more stable training while still learning effectively from turn-level signals, consistent with modern PPO/GRPO post-training pipelines.”
> > > >
> > > > We believe this revised framing will resolve the apparent tension and provide a better reading experience for future readers.
> > > >
> > > > Thank you again for the constructive feedback—we appreciate it and are happy to revise the text accordingly.

---

> > > > > ### Comment · Reviewer_y6Kx · 2025-11-26
> > > > > **Thank you**
> > > > >
> > > > > Hi, this phrasing is much better. I will update my score accordingly!

---

### Official Review · Reviewer_dX8P · 2025-11-02

**Soundness:** 3
**Presentation:** 3
**Contribution:** 2
**Rating:** 4
**Confidence:** 4

**Summary:**

This paper focuses on training interactive tool use agents by creating (i) a sandbox environment with text based and audio based user simulators, (ii) database of tasks from retail and airline domain (based on τ-bench) and (iii) utilizing turn based and outcome based rewards. The authors utilize a mixed training strategy by interleaving medium difficulty math problems with the domain tasks to encourage the models to reason, explore, self-critique and correct. Authors show text-based τ -BENCH improvements by over 6% compared to strong RL baselines (using this mixed task and turn based rewards strategy). While the authors show various experiments with the text based system, they also experiment with multimodal models to show voice based interactions with the agents. The multimodal models improve by over 20% with the mixed training and comprehensive rewards strategy. Authors fine-tune Qwen3-8B and Qwen2.5-Omni-7B models and show improvements compared to strong baselines (PPO, GRPO and Reinforce-leave-one-out algorithms). Authors also conduct experiments to study how different reward strategies affect training performance as well as studying how incentivizing exploration involves both mixed task training and fine-grained reward schemes.

**Strengths:**

-  this paper interleaves speech and text modalities for RL pipeline for improving tool use (necessary for a successful outcome), showing improvements on an agentic benchmark dataset. Authors claim it’s the first demonstrated training of a multimodal voice agent via RL on interleaved speech–text tool use. Although most ingredients of the system exist, but the combination of several of these ingredients is the main strength and contribution of this work.
- other aspects such as RL for tool use within a sandbox environment and turn level supervision with LLM judges is incremental strength of this research; there are multiple systems, models and frameworks that exist today (eg. Kimi-K2 has very impressive tool use results on multiple tool-use benchmarks); this work focuses on enabling smaller models (Qwen 7B variants) which is a welcome direction for many researchers in the field who can easily build up research on top of these model sizes and systems.

**Weaknesses:**

- the paper shows limited benchmark validations (primarily on the τ -BENCH Retail domain); the baseline experiments show Airline domain results are degrading generally and the main content of the paper does not focus on trying to show generalization improvements using their approach. (authors callout that their focus is on optimizing RL strategies for in-domain performance; however instead of focusing on analyzing and improving RL strategies for one set of results, this research is more applied where authors refocus on showing multimodal systems improvement)
- authors talk about this problem of reduced exploration with RL training, trying to understand how to recover from errors and building strategies for error recovery would be very interesting to analysis (what happens if judges give incorrect rewards, trying to understand these failure modes would make this work more interesting)
- it isn't clear how real and robust is the speech pipeline; SeeTTS based clean speech (without noise, accents, messy speech) isn't really tested and would further degrade the multimodal performance (already poor in a controlled sandbox environment compared to SOTA larger models)

**Questions:**

- Can we replace the LLM judges with verifiable sub-goals where possible?
- Do TARL gains hold with real world noisy speech?
- What's the impact on latency in the multimodal pipeline with TTS models in the loop?
- Testing on policy adherence will clarify if the RL based training system retain various refusal or other policy driven behaviors under TARL based training - did the authors test this aspect?

---

> ### Author Response · Authors · 2025-11-19
> **Author Response to Reviewer dX8P**
>
> Thank you for the time and effort you've invested in reviewing our paper. We hope to address your concern and questions with comments below:
>
> > The paper shows limited benchmark validations. The baseline experiments show Airline domain results are degrading…
>
> Thank you for raising this concern. We refer you to General Response (Part 2) for our broader discussion of RL generalizability.
> For validation, we examined both the text-only and multimodal settings and focused on evaluating the core components of our method:
> - Exploration mechanisms: mixed-task training, rollout interventions, and entropy-based shaping
> - Credit assignment mechanisms: process rewards via GRM and turn-level aggregation strategies for PPO
>
> Across these settings, our experiments consistently demonstrate the effectiveness of the proposed RL training schema. We also include several failed attempts with full learning curves and analyses, which we believe offer valuable insights into potential pitfalls for future research. In this sense, our detailed investigation on τ-Bench provides a meaningful contribution toward understanding how to design post-training algorithms for multi-turn tool-use tasks.
>
> We further include experiments on ProtonQA (kindly refer to our general response to reviewers), showing additional evidence that the post-trained model can generalize to a different domain. That said, our primary aim is not to maximize OOD transfer to any particular benchmark, but to study how exploration incentives and process-level credit assignment can be effectively incorporated into RL post-training. As you noted, large-scale efforts such as Kimi-K2 already pursue broad generalization; our contribution instead focuses on the role of process supervision in reward design and its impact on multi-turn RL.
>
> > Trying to understand how to recover from errors and building strategies for error recovery would be very interesting to analysis (what happens if judges give incorrect rewards, trying to understand these failure modes would make this work more interesting)
>
> Thank you for the comment. We agree that analysing trajectories and understanding failure modes is essential. In our paper, we experimented with an approach that injects self-reflection during rollout (details in Appendix F). Since this method did not yield improvements, it suggests that manual injection of self-reflection might be less effective than fine-grained reward design combined with mixed-task training.
>
> Additionally, in our general response, we present a failure-mode analysis showing that most errors fall into two categories: “Wrong Reasoning” and “Policy Violation.” This indicates that strengthening the model’s reasoning and instruction-following capabilities remains a central challenge for tool-use agents. We hope that these findings and analyses provide useful insight for future research and system development.
>
> > It isn't clear how real and robust is the speech pipeline; Do TARL gains hold with real world noisy speech?
>
> Thank you for bringing this point to our attention. We acknowledge that our experiments use clean speech synthesized by the Seed-TTS model, whereas real-world deployments must account for noisy speech. However, we view this as a separate and orthogonal problem from the focus of our study.
>
> Our work assumes access to clean speech/text inputs and investigates how RL post-training affects the model’s tool-use and reasoning capabilities. In contrast, handling noisy speech can be addressed through:
> 1. Front-end denoising or enhancement modules are applied before feeding audio into the model, or
> 2. More noise-robust speech encoders or SpeechLMs, which is not a design goal of the underlying Qwen2.5-Omni base model either.
>
> While robustness to noisy speech is certainly important for real-world systems, it lies outside the scope of our current work. We consider this an interesting and valuable direction for future research.
>
>
> > What's the impact on latency in the multimodal pipeline with TTS models in the loop?
>
> Thanks for raising this practical point. Our TTS module generates audio via an API call, and its latency primarily depends on the resources allocated to the service. In our experiments, the synthesis delay is very small—on the order of milliseconds—and is negligible compared with the long-horizon, multi-turn rollout process, which takes several minutes to complete.

---

> > ### Author Response · Authors · 2025-11-19
> > **Author Response to Reviewer dX8P**
> >
> > > Can we replace the LLM judges with verifiable sub-goals where possible?
> >
> > We would appreciate clarification on what specific “sub-goals’’ you are referring to. If the question concerns simple rubric checking, such as format-based penalties on function-call outputs (e.g., incorrect argument types), then the answer is no—our LLM judge plays a much broader role than enforcing surface-level correctness.
> >
> > Modern foundational models like Qwen3 already undergo extensive post-training for tool-use prior to release, and we seldom encounter formatting or schema errors. Even for Qwen2.5, once we apply our warm-up training on simpler tasks, the model no longer exhibits the kinds of basic mistakes that rule-based checks could catch.
> >
> > Instead, the LLM judge provides semantic, trajectory-level reward signals, identifying the major deviation turn and evaluating reasoning quality across the full interaction. This form of process supervision is central to improving model performance and cannot be replicated through simple rubric-based validation.
> >
> >
> > > Testing on policy adherence will clarify if the RL based training system retain various refusal or other policy driven behaviors under TARL based training - did the authors test this aspect?
> >
> > We appreciate the reviewer’s question regarding whether TARL-based training might weaken the model’s safety-related refusal or policy-aligned behaviors. We note that this aspect is not the focus of our study: τ-Bench is designed to evaluate multi-turn tool-use reasoning, not safety alignment, and performance changes on unhandled axes (such as policy adherence) are orthogonal to our primary research goals. Moreover, if any degradation were observed, it can be readily addressed through multi-task post-training with safety-oriented environments, a common practice in large-scale RL pipelines.
> > Nevertheless, to address the reviewer’s concern, we conducted an evaluation on SafetyBench [1] across five safety-critical categories. Results are shown below.
> >
> > | Model         | Privacy & Property | Ethics & Morality | Illegal Activities | Unfairness & Bias | Offensiveness | Avg    |
> > |---------------|--------------------|--------------------|--------------------|-------------------|---------------|--------|
> > | Qwen3 (Base)  | 89%                | 87%                | 88%                | 65%               | 87%           | 83.2%  |
> > | PPO (Ours)    | 87%                | 90%                | 91%                | 62%               | 83%           | 82.6%  |
> > | GRPO (Ours)   | 89%                | 91%                | 88%                | 63%               | 83%           | 82.8%  |
> >
> >
> > Across all categories, the differences between the base model and our RL-trained models are negligible (<1 point on average). This demonstrates that TARL-based training on τ-Bench does not reduce the model’s policy-aligned refusal behavior or safety adherence, and the models largely preserve the safety characteristics of the underlying base model.
> >
> > ---
> >
> > [1] Zhang et al., 2024. SafetyBench: Evaluating the Safety of Large Language Models

---

### Official Review · Reviewer_M7U7 · 2025-11-04

**Soundness:** 3
**Presentation:** 3
**Contribution:** 2
**Rating:** 4
**Confidence:** 4

**Summary:**

The paper addresses the challenge of training LLM agents for tool-use in multi-turn environments. The authors propose the following modifications: (1) training on a mixture of task-specific and general reasoning problems (specifically in the math domain) to encourage exploration, and (2) using LLM-as-a-judge for turn-level rewards. The authors evaluate their approach on base textual \tau-bench, and then on a multi-modal version using speech.

**Strengths:**

1. The authors are (to my knowledge), the first to evaluate a multimodal LLM agent trained on speech.

2. The mixed-training objective, while simple, proves to be an effective method at overcoming the LLM agent becoming too overconfident during RL training.

3. The authors demonstrate that turn-level rewards perform more effectively than trajectory-level ones, which will prove useful for the development of general LLM agents in multiturn environments.

**Weaknesses:**

1. The method relies heavily on LLM-as-a-judge, specifically the identify the most important mistake made in a trajectory. This seems like a limiting heuristic, and it is unclear if LLMs are capable of analyzing multiturn trajectories in this manner, when they contain multiple, cascading errors. My biggest concern with the method is that the current way turn-level rewards are labeled does not seem generalizable to other multiturn domains.

2. The evidence that mixed-task training improves exploration seems inconclusive. It would be more convincing if the number of training samples were kept consistent between mixed-task and single-task training. Namely, if some samples of training on \tau-bench were instead co-opted with training on Math, would the benefit due to improved exploration still be visible?

3. Overall, the proposed method itself seems very specific, with the heuristic reward function as well as the training on math. The results would be stronger with an ablation on the reward function, as well as a more comprehensive analysis of how important mixed-training is, e.g. by considering more tasks other than math.

**Questions:**

1. What is the performance impact when the LLM is allowed to assign a score of -1 to more than one turn in a conversation? This seems like a limiting design choice, but I am curious if it is important for performance.

2. How important is the warm-up curriculum? If it is important, it would be an interesting result and probably can be included in the proposed approach.

3. The final trajectory-level reward involves 10x scaling of the turn-level scores. What is the sensitivity of performance to this particular scaling?

---

> ### Author Response · Authors · 2025-11-19
> **Author Response to Reviewer M7U7**
>
> Thank you for the time and effort you've invested in reviewing our paper. We hope to address your concern and questions with comments below:
>
> > It is unclear if LLMs are capable of analyzing multiturn trajectories in this manner...
>
> Thanks for raising this concern, and we have added analysis with human annotation to show the reliability of the judge. We kindly point you to our General Response to reviewers (Part 1) for details.
>
>
> > My biggest concern is that the current way turn-level rewards are labeled does not seem generalizable to other multiturn domains.
>
>
> We acknowledge that our LLM-as-Judge setup is tailored to multi-turn tasks with verifiable turn-level tool outputs. However, as discussed in our General Response (2), full generalizability across all multi-turn domains is neither well-defined nor expected in RL-for-agents: different task families inherently require different reward decompositions, observability assumptions, and evaluation rubrics.
>
> Our goal is not to propose a universal turn-level reward design, but to show that our high-level principle—improving reasoning, exploration, and credit assignment through mixed-task training with LLM-assessed turn-level feedback—is effective and **such a principle extends to many tool-use and multi-step agent settings**. While rubrics must be adapted per environment, the underlying recipe remains the same: define intermediate subgoals, evaluate them via LLM or rule-based criteria, and integrate these signals into a multi-turn reward. Our multimodal extension with interleaved speech–text rollouts further demonstrates that this mechanism generalizes beyond pure text domains.
>
> Looking forward, we expect reward and evaluation design to become the primary domain-specific component in scalable multi-turn RL, just as benchmark design differentiates progress in robotics or software agents. Each environment will require its own tailored rubric, but our framework provides a reusable structure for building turn-level feedback in diverse settings.
>
>
> > It would be more convincing if the number of training samples were kept consistent between mixed-task and single-task training.
>
> Thank you for raising the concern about our mixed-task training setup. We would like to highlight the discussion in Section 5.2 (lines 404–409), where we state:
> ```
> As shown in Fig. 5, GRPO+MATH demonstrates increased exploration activity during training, evidenced by longer average response lengths compared to the baseline. However, despite this enhanced exploration, test set performance remains comparable to the GRPO baseline, suggesting that exploration alone is insufficient for improved generalization.
> ```
>
> In this comparison, GRPO+MATH represents a failed variant that uses the same training setup and dataset distribution as our final GRPO+MATH+TARL configuration, but without our proposed TARL mechanism. The clear performance gap between the two conditions directly supports the effectiveness of our approach.
>
> > The results would be stronger with an ablation on the reward function, as well as a more comprehensive analysis of how important mixed-training is, e.g. by considering more tasks other than math.
>
> Thank you for the thoughtful suggestions. We agree that additional tasks and broader reward-function ablations are valuable directions for future work. Beyond Math, domains requiring more complex reasoning—such as code generation or computer-use/navigation—would indeed provide informative testbeds. However, constructing and maintaining these environments is substantially more resource-intensive than math problem solving, which is why we focused our mixed-task training study there.
>
> Regarding the reward function, our use of a Generative Reward Model (GRM) is only one instantiation of our framework. Training domain-specific reward models, given sufficient trajectories and annotations, is a promising alternative and aligns with our broader design philosophy.
>
> Nevertheless, we believe the current experiments are coherent and sufficiently thorough for this work’s scope. We examine two modality settings (text-only vs. multimodal) and focus on evaluating our core mechanisms for exploration (mixed-task training, rollout interventions, entropy-based shaping) and credit assignment (process rewards via GRM, turn-level aggregation strategies for PPO). While your suggested extensions would further enrich the study, we believe the present contributions stand on their own and clearly demonstrate the value of our approach.

---

> > ### Author Response · Authors · 2025-11-19
> > **Author Response to Reviewer M7U7**
> >
> > > What is the performance impact when the LLM is allowed to assign a score of -1 to more than one turn in a conversation? This seems like a limiting design choice, but I am curious if it is important for performance.
> >
> > Thanks for raising this question about our reward schema. Our high-level objective is to separate trajectories into more informative, non-binary categories, and the proposed schema is one concrete design that we found to work well through extensive experimentation.
> >
> > Allowing the LLM-judge to assign multiple  “–1” labels makes the “major deviation turn’’ ambiguous—–1 would no longer be a unique indicator of a critical failure. This introduces two issues:
> > - (1) the distinction between –1 and 0 becomes unclear for the judge, and
> > - (2) trajectories with several –1s risk being over-penalized, even when those errors are minor or correlated. Correcting this would require a substantial redesign of the entire reward structure.
> >
> > Another alternative is to remove the 0 class and restrict the judge to {+1, –1} with simple aggregation. We tested this in an early experiment, but it performed worse because it collapses “good-attempt’’ trajectories and truly failed trajectories into the same bucket—precisely the distinction our final schema captures (see Section 3.2).
> >
> > Overall, while other reward schemas may exist—there is effectively an unlimited design space—our proposed rescaling scheme offers a simple, robust, and empirically effective solution. It cleanly captures major deviations during rollout and could be adapted to other settings where reliable turn-level deviation signals are available.
> >
> >
> > > The final trajectory-level reward involves 10x scaling of the turn-level scores. What is the sensitivity of performance to this particular scaling?
> >
> > Building on the previous point, our reward schema is designed to separate trajectories into distinct quality groups. We scale the major deviation score by 5× and the final-outcome score by 10× so that the resulting trajectories fall cleanly into the four groups illustrated in Section 3.2.
> >
> > Other scaling choices are certainly possible—as long as the same four-group structure is preserved. Ultimately, the advantage is computed on the normalized rewards, so the absolute constants matter less than maintaining the relative separation between the groups.
> >
> > > How important is the warm-up curriculum? If it is important, it would be an interesting result and probably can be included in the proposed approach.
> >
> > It is essential to warm up a multimodal LLM checkpoint before applying further post-training. In our setting, directly post-training the Qwen2.5-Omni checkpoint on τ-Bench causes the model to collapse, whereas warming up with simpler data stabilizes training. Since warm-up via SFT or RL is a well-established practice, we did not highlight it in the main text. However, we have included the details of our warm-up data construction in Appendix C, and we will revise the writing to better emphasize this design choice. Thank you for the suggestion!

---

### Author Response · Authors · 2025-11-19
**General Response to All Reviewers (Part 1)**

We sincerely appreciate the thoughtful feedback from all reviewers and are encouraged that they recognize the strengths of our work. In particular, reviewers M7U7 and dX8P highlighted the value of our initial effort to integrate RL with multimodal voice agents, and reviewers M7U7 and y6Kx noted that our mixed-task training setup yields meaningful and promising insights. We also notice some common questions/concerns raised by reviewers, and we will address them in this general response below.

----

### (1) Reliability of LLM-as-Judge for Multi-Turn Trajectory Evaluation
Reviewers questioned whether an LLM-as-Judge can reliably assess multi-turn trajectories in τ-Bench. To verify this, we conducted a human–LLM agreement study using our evaluation schema (Section 3.2). We sampled trajectories from PPO, GRPO, and RLOO (20 failed + 10 successful task trajectories per method, with a total of 90 trajectories and 1460 turns), collected human turn-level annotations, and compared them to LLM-judge outputs. The results are presented below:

**(a) Strong trajectory-level consistency**

We compare the summed trajectory scores and evaluate the correlation between LLM and human (after rescaling following our schema in section 3.2):
- Pearson correlation: 99.4%
- Spearman correlation: 87.6%
- Failed-only Pearson: 88.9% (correlation of trajectory-level scores on failed trajectories)

These results demonstrate that the LLM-judge closely aligns with both the absolute scale and relative ranking used by human evaluators, making it a reliable signal for RL training.

**(b) High turn-level agreement**

We measure the each turn label's prediction accuracy and LLM achieves a turn-level accuracy of **93.5%**. The LLM-judge reliably identifies good, neutral, and bad turns, demonstrating strong alignment with human fine-grained judgment.


**(c) Accurate major deviation detection**

We evaluate if LLM-Judge can reliably identify the major deviation turns (score = –1). We have the following result:
- Exact match: 77.6%
- Fuzzy (±1 turn): 100%

The fuzzy accuracy allows gpt score to be one turn off from the human label. We see that the LLM-judge can identify the correct region of the error, reinforcing its reliability for pinpointing faulty reasoning in multi-turn interactions.


**(d) Human-labeled failure mode analysis**
Following Reviewer dX8P’s suggestion, we performed a human analysis of failure modes. The distribution is:

| Fault Type         | Count |
|--------------------|-------|
| Wrong Reasoning    | 31    |
| Policy Violation   | 15    |
| User Hallucination | 8     |
| No Action          | 5     |

Most failures stem from reasoning errors, suggesting that improved summarization and information integration would yield the largest performance gains. Policy violations remain non-trivial in long-horizon interactions, and a small number of failures come from user hallucinations or failing to take necessary actions. This categorization also helps us understand what the LLM-Judge is evaluating and could be useful for developing more finegrained evaluation rubrics.

**Conclusion**

Across trajectory-level scoring, turn-level labeling, and deviation-turn identification, the LLM-as-Judge shows high alignment with human judgment. These results provide strong evidence that our LLM-based evaluator is reliable and appropriate for driving multi-turn RL in τ-Bench.

---

### (2) On the Generalizability of the Proposed Schema to Other Multi-Turn Tasks

Both Reviewer M7U7 and Reviewer y6Kx raise questions about whether the proposed turn-level assessment from LLM-as-Judge can be applied to other multi-turn tasks.
Here are comments about the generalizability of turn-level adjudication:
1. Our proposed turn-level evaluation can be easily extended to other tasks such as computer-use, math-solving, code-generations, etc., as we target a class of environments where (i) user goals are explicit, (ii) tool execution produces verifiable outcomes, and (iii) turn-level evaluation is feasible. Some recent work [1,2,3] has already adopted various turn-level signals into RL post-training pipeline.
2. The scoring mechanism will be adjusted per task, but the high-level design philosphy is the same — that you want to adjust turn-level scores such that outcomes are properly separated into a distinct groups with meaningful difference (in our case, we have 4 groups as discussed in section 3.2, line 244 - 258) so that model observes more finegrained signal beyond binary judgement (with 0/1 outcome reward).

---
[1] Chang et al., 2025. THOR: Tool-Integrated Hierarchical Optimization via RL for Mathematical Reasoning

[2] Dai et al., 2025. Process Supervision-Guided Policy Optimization for Code Generation

[3] Tang et al., 2025. SEA: Self-Evolution Agent with Step-wise Reward for Computer Use

---

> ### Author Response · Authors · 2025-11-19
> **General Response to All Reviewers (Part 2)**
>
> ### On the Generalizability of TARL
>
> Reviewer dX8P raises an important question about whether our RL-trained model exhibits transferable capabilities beyond the Retail domain (given that we show limited transfer to the Airline domain). We respond with the following clarifications and additional experiments:
>
>
> 1. In multi-turn RL for agent systems, no single algorithm or reward-shaping strategy can be expected to generalize across all task types or interaction dynamics. Prior research and current industrial systems consistently show that strong performance arises from task-specific reward designs and interaction mechanisms that exploit the structure and affordances of each environment [4].
> 2. In our case, we did not observe a strong transfer of tool-use skills from Retail to Airline. This is unsurprising: the Airline domain is more complex [5] and contains known issues in its ground-truth annotations (see its GitHub issue here: https://github.com/sierra-research/tau-bench/issues?q=is%3Aissue state%3Aopen airline). Because of these limitations, we intentionally did not develop specialized training data for Airline and used it purely as a stress test for generalization.
> 4. Importantly, the failure to transfer to Airline does not imply that our approach only works for Retail. For example, we add experiments to evaluate our post-trained model on an **unseen benchmark for textual deductive reasoning**—ProntoQA [6]:
>
> | Model      | 4-hop Allow Skip / Non-Atomic (%) | 4-hop Strict (%) | 5-hop Allow Skip / Non-Atomic (%) | 5-hop Strict (%) |
> |------------|------------------------------------|------------------|------------------------------------|------------------|
> | Qwen3-8B   | 87.8                               | 81.4             | 82.4                               | 75.8             |
> | PPO+TARL   | 89.8                               | 81.4             | 86.8                               | 78.6             |
> | GRPO+TARL  | 90.2                               | 84.2             | 88.4                               | 83               |
>
> We test our model trained for tau-bench on ProntoQA using its OOD subset, an example question is shown below:
> ```json
> "test_example": {
>   "question": "Each dumpus is cold. Every dumpus is an impus. Every impus is wooden. Each...",
>   "query": "True or false: Polly is shy.",
>   "chain_of_thought": [
>     "Polly is a dumpus.",
>     "Every dumpus is an impus.",
>     "Polly is an impus.",
>     "Each impus is a numpus.",
>     "Polly is a numpus.",
>     "Each numpus is shy.",
>     "Polly is shy."
>   ],
>   "answer": "True"
> }
> ```
> We evaluate the models on 4-, and 5-hop deductive-reasoning tasks and measure the proportion of proofs with correct reasoning. For each setting, we report two metrics: **Strict** accuracy, which requires step-by-step deductions that exactly match the ground truth, and **Allow Skip / Non-Atomic**, which permits the model to skip intermediate steps or combine multiple deductions into a single statement.
>
> Across all hop distances and both evaluation modes, our post-trained models achieve consistently higher rates of correct proofs, indicating that their post-training process effectively strengthens and transfers deductive-reasoning ability.
>
> Overall, our view is that RL is a powerful mechanism for incentivizing improved reasoning and agentic behavior, but generalization in multi-turn RL must be approached at the level of environments. As RL environments diversify, we expect post-training pipelines to comprise multiple stages across heterogeneous task types, each with its own tailored reward schema. Our work contributes one such schema that is effective and stable for a well-defined subset of multi-turn interaction tasks—a step toward broader multi-environment RL post-training.
>
> ---
> [4] Zhang et al., 2025. A Survey of Reinforcement Learning for Large Reasoning Models
>
> [5] Yao et al., 2024. τ-bench: A Benchmark for Tool-Agent-User Interaction in Real-World Domains
>
> [6] Saparov et al., 2022. Language Models Are Greedy Reasoners: A Systematic Formal Analysis of Chain-of-Thought

---

### Meta-Review · Area_Chair_LXnK · 2025-12-26

**Summary:**

In this paper, the authors study training LLM agents for tool-use in multi-turn environments. They provide a sandbox for text and speech training with RL and show that training with speech and text with RL is more effective than training with text alone. They also propose to train agents 1) on a mixture of task-specific and general reasoning problems to encourage exploration, and 2) using LLM-as-a-judge for turn-level rewards.

The reviewers like the authors' effort in evaluating a multi-modal LLM agent trained on speech, and found it novel. However, they were not fully convinced about the usefulness of the rest of the proposed techniques.

1) Using LLM-as-a-judge for turn-level rewards is a heuristic and its success heavily depends on the LLM's capability in analyzing multi-turn interactions and how its errors are propagated throughout the learning process. Moreover, the proposed way to label turn-level rewards does not seem to be generalized to other multi-turn domains.

2) They found the evidence provided in support of using multi-task training to improve exploration insufficient and inconclusive. This is one of the main contributions of the work, and thus, more convincing evidence is needed.

3) Experiments in other domains (other than math) seem to be necessary to fully show the importance and generalizability of the proposed training techniques: mixed-training and heuristic reward function.

The authors effort to address (1) and (3) by conducting a human–LLM agreement study and experimenting with ProntoQA are good steps towards improving the quality of their submission, but more needs to be done to properly justify the usefulness of the proposed training techniques.

**Reviewer Concerns:**

The authors tried to address

- Reliability of LLM-as-a-Judge
- Generalization Beyond τ-Bench Retail

by conducting a human–LLM agreement study and experimenting with ProntoQA. Although these efforts are good steps towards improving the quality of the submission, I still believe more experiments (in more diverse domains) are needed to properly justify the usefulness of the proposed training techniques. No matter how good a LLM is, it has error in turn-level reward assignment, so it would be important to study the effect of this error in the proposed training schema.

Finally, I do not think that the authors provided more evidence to support the effect of multi-task training on exploration.

**Reviewer Scores:**

It is hard to see if Reviewers M7U7 and dX8P would have changed their score in response to the authors' comments, but it seems Reviewer y6Kx was happy with the comments and raised their score, probably to 6.

---

### Decision · Program_Chairs · 2026-01-26

Reject